# Optimisation of Segregation Distances between Electric Cable Bundles Embedded in a Structure

Jérôme Morio [1,*], Isabelle Junqua [2], Solange Bertuol [2] and Jean-Philippe Parmantier [2]

1 ONERA/DTIS, Université de Toulouse, 31055 Toulouse, France
2 ONERA/DEMR, Université de Toulouse, 31055 Toulouse, France; isabelle.junqua@onera.fr (I.J.); solange.bertuol@onera.fr (S.B.); jean-philippe.parmantier@onera.fr (J.-P.P.)
* Correspondence: jerome.morio@onera.fr

**Abstract:** This paper presents the optimisation of the segregation distance between two electric cable bundles installed in an aircraft structure under electromagnetic compatibility constraints. We first describe the problem formulation where a probabilistic constraint has to be verified during the optimisation process. To overcome the nonlinearity of the constraint function and guarantee the algorithm convergence, we propose a joint approach between Monte Carlo sampling and a Kriging surrogate to estimate the optimum distance with a low computational cost. This methodology was tested on a realistic use-case of distance segregation between cable bundles.

**Keywords:** electromagnetic susceptibility; segregation distances; uncertainty quantification; reliability-based design optimisation

## 1. Introduction

To avoid electromagnetic (EM) compatibility problems, the allocation of elementary electrical cables in a bundle and the installation of cable harnesses in structures must be controlled and respect the segregation rules set today by the integrators. To this extent, in aeronautics, the wiring, named the Electrical Wiring Interconnection System (EWIS), is now considered as a system as a whole, which must fulfil the Certification Specifications for Large Aeroplanes (CS25 subpart H).

Consequently, these constraints must be considered early in the upstream phase of a program, when cable harnesses are defined while most input data are not completely mastered or are even unknown (as cable lengths, relative location of harnesses, location versus the structure, etc.). This is the reason why only some routes in which harnesses can be grouped to run together are predefined. With respect to the signal nature and the electrical power on cables, distances between routes are imposed to secure the EM inter-compatibility between cables. Moreover, once these installation rules are defined, it is necessary to validate them for certification purposes. However the update of the computer-aided design (CAD) mock-up because of modifications of the structure or the installation of the harnesses in the structure can lead to deviations from the installation rules, by reducing for example the distance between two harnesses to a distance lower than the recommended segregation distance. In this case, one must be able to evaluate the potential risk brought by this deviation in order to justify it and to obtain the derogation.

The objective of this article was thus to develop a modelling strategy to evaluate the optimal segregation distance between two electric cable bundles connecting two equipment, which will ensure their EM inter-compatibility. Therefore, we wanted here to minimise the distance between two bundles to save space for the installation of harnesses provided that the electromagnetic compatibility (EMC) constraints are respected with a given probability. We show in this article that this problem is in fact similar to a stochastic optimisation with probabilistic constraints [1]. Different methods have been proposed for this range of problems, and we particularly concentrate here on surrogate models and stochastic sampling.

Besides, a recent review of this class of algorithms for optimisation under probabilistic constraint was performed with application to reliability in [2]. In this article, we also focus on the joint approach between Monte Carlo and Kriging surrogate models where the Monte Carlo sampling enables estimating the probability that the EM constraint is fulfilled and the Kriging model mimics the objective and the constraint function in an augmented space. The difficulty lies here in the nonlinearity of the constraint function, notably at high frequency, which can make the surrogate model unsuitable. Due to industrial constraints to provide a converged result in a reasonable time, whatever the use-case, we propose here to switch from the Kriging surrogate model with Monte Carlo sampling to classical Monte Carlo sampling when the surrogate model is not efficient. This question has been addressed in the Clean Sky 2 ANALYsis Statistical Techniques in aeronautics (EMC ANALYST) project (CFP07 GA 821128), gathering two Italian (Ingegneria dei Sistemi (IDS) and University of L'Aquila) and two French (AxesSim and ONERA) partners to answer the topic manager's (Safran Electrical & Power) requirements.

In this article, we first describe the problem of segregation distance between cable bundles in Section 2. In Section 3, we show that this issue is in fact an optimisation problem under a probabilistic constraint where the constraint function is nonlinear due to the physics. Section 4 is dedicated to the statistical process we propose to solve this optimisation, and the last section develops the obtained results on the considered use-case.

## 2. EMC between Two Bundles

### 2.1. Electric Cable Bundle Configuration

Let us consider two electric cable bundles $A$ and $B$ composed respectively of $n_A$ and $n_B$ elementary electric cables. Both cable bundles are parallel with each other along the whole path. We also assumed that the following characteristics are known for any elementary cable $i \in [\![1, n_A]\!]$ of bundle $A$ and $j \in [\![1, n_B]\!]$ of bundle $B$, as presented in Figure 1:

- The input impedance of the equipment at both extremities, denoted as $Z_{A,NE,i}$-$Z_{A,FE,i}$ and $Z_{B,NE,j}$-$Z_{B,FE,j}$;
- The source generators, applied at one extremity of both bundles and defined as voltage generators, $V_{0,A,j}/V_{0,B,j}$ or current generators $I_{0,A,j}/I_{0,B,j}$;
- The EM susceptibility (EMS) thresholds of the equipment at the extremity opposite the source application ($S_{A,i}$ and $S_{B,i}$). They are intrinsic to each equipment and are measured in specific conditions of installation defined in the EMC standards. They can be expressed in terms of voltages or currents;
- The cable length $l$ (assumed to be identical for both harnesses $A$ and $B$);
- The cable heights of both harnesses, $h_A$ and $h_B$, over the reference ground plane.

The source generators and EMS thresholds are given in the frequency domain.

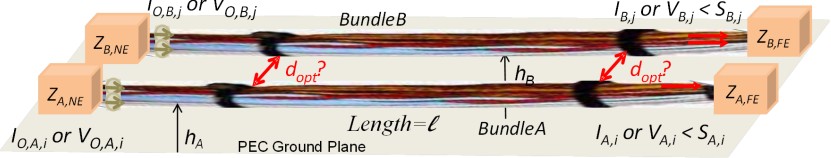

**Figure 1.** Analysis of the EMC of 2 parallel cable bundles.

The currents and voltages on the elementary wires of both bundles, $I_{A,i}/V_{A,i}$ and $I_{B,j}/V_{B,j}$, are induced by the useful signal imposed by voltage/current generators on the elementary wires, crosstalk between the wires in the same bundle, and crosstalk between bundles $A$ and $B$.

The consistency between the EMS thresholds and source generators is supposed to be fulfilled for all elementary cables. Furthermore, we also assumed that both bundles $A$ and $B$ have been correctly designed, which means that they are EM self-compatible. In other words, the signals carried by a cable of a given bundle will not induce significant interference on their neighbouring cables inside the bundle.

### 2.2. Inter-Compatibility Criterion

The EM susceptibility of an equipment input, $S_{A,i}$ ($S_{B,j}$), can be defined as a voltage or a current and as a limit below which the equipment operates correctly. As an example, if $S_{A,i}$ is defined as a threshold current (respectively voltage), the magnitude of the induced current $I_{A,i}$ (respectively voltage $V_{A,i}$) must be compared to $S_{A,i}$. This is the reason why we introduced $O_{A,i}$ and $O_{B,j}$ as either the magnitude of the currents ($I_{A,i}$ or $I_{B,j}$) or the magnitude of the voltages ($V_{A,i}$ or $V_{B,j}$) following the nature of the susceptibility thresholds ($S_{A,i}$ or $S_{B,j}$). $O_{A,i}$ and $O_{B,j}$ induced on each elementary cable $i$ of bundle $A$ and $j$ of bundle $B$ by the source generators depend on the frequency $f$, the segregation distance $d \in [d_{\min}, d_{\max}]$, the coupling length $l \in [l_{\min}, l_{\max}]$, the heights $h_A \in [h_{\min}, h_{\max}]$, $h_B \in [h_{\min}, h_{\max}]$, and finally, on the electric characteristics of all elementary cables in each bundle. The computation of $O_{A,i}$ and $O_{B,j}$ can be performed with a computer code named CRIPTE [3,4]. This code developed at ONERA for more than 20 years is dedicated to the evaluation of electromagnetic interferences induced on multiconductor cable networks, for typical applications as lightning indirect effects and internal EMC problems (crosstalk). It is based on a topological formalism associated with the transmission line theory generalised to multiconductor networks. The code is mature enough to model cable networks of industrial complexity such as engine cable architectures [5].

The next step consists of defining a mathematical scalar criterion to determine if the two bundles $A$ and $B$ are EM compatible. For this purpose, we propose to compute a $G$ function defined as:

$$G(d, l, h_A, h_b) = \max_{f \in F, \, i \in [\![1, n_A]\!], \, j \in [\![1, n_B]\!]} \left( \frac{O_{A,i} - S_{A,i}}{S_{A,i}}, \frac{O_{B,j} - S_{B,j}}{S_{B,j}} \right) \tag{1}$$

The event $G(d, l, h_A, h_B) < 0$ means that both bundles $A$ and $B$ are EM compatible, and EM immunity is ensured for all equipment and, conversely, if $G(d, l, h_A, h_B) > 0$. Indeed, if $G < 0$, it means that $O_{A,i} < S_{A,i}$, $\forall i$ and $O_{B,j} < S_{B,j}$, $\forall j$. For instance, if $S_{A,i}$ is defined in terms of current, it means that $O_{A,i}$ are the currents induced on bundle $A$, and these currents are lower than the threshold acceptable by the equipment. Thus, we deduced that the equipment will operate correctly, and we can conclude that $A$ and $B$ are compatible.

The normalisation of the induced currents/voltages to their corresponding susceptibility thresholds enables obtaining a dimensionless $G$ criterion, which is independent of the nature of the susceptibility thresholds (themselves defined as currents or voltages). Let us remark also that $G$ in Equation (1) covers the whole frequency band $F$. The computation time to obtain one evaluation of G with CRIPTE is about 1 min for a thousand frequencies and bundles of about thirty elementary conductors for a given set of $(d, l, h_A, h_b)$.

### 3. Segregation Distance Optimisation Problem with EMC Constraints

The objective in this article was to determine the smallest value of the distance $d$, denoted as $d_{opt}$, under the constraints that:

1.  $d_{opt} \in [d_{\min}, d_{\max}]$;
2.  $G(d_{opt}, l, h_A, h_B) < 0$ for any $l$, $h_A$, and $h_B$;
3.  For every value $d > d_{opt}$ and for any $l$, $h_A$, and $h_B$, $G(d, l, h_A, h_B) < 0$.

This definition of the optimal segregation distance $d_{opt}$ is in fact very conservative. As $G$ is not a monotonic decreasing function with $d$, our objective was to find the maximum of different minima $d$ that satisfy the constraints. For a given use-case, it may be possible to find a distance $d'$ lower than $d_{opt}$ that will be EM compatible, but if such a distance $d'$ exists, by definition, we can also find a distance $d''$ with $d' < d'' < d_{opt}$ such that $d''$ does not lead to an EM-compatible configuration. Moreover, these conditions are computationally hard to verify since the $G$ function has to be evaluated for every combination of $l$, $h_A$, and $h_B$ at a given distance $d$. We thus propose to replace these constraints with probabilistic ones. For this purpose, we now consider that the length $l$ and the heights $h_A$ and $h_B$ can be represented by three random variables. We thus define three independent uniform random

variables $L$, $H_A$, and $H_B$, respectively, between the intervals $[l_{\min}, l_{\max}]$ and $[h_{\min}, h_{\max}]$. The associated random vector $(L, H_A, H_B)$ is denoted as $\mathbf{X}$ with a probability distribution $f_{\mathbf{X}}$. The probability $\mathbb{P}(G(d, \mathbf{X}) < 0)$ defines the probability that bundles $A$ and $B$ are EM compatible at a distance $d$. Then, the previous minimisation problem can thus be rewritten in the following way as we seek to evaluate $d_{opt}$ with:

$$d_{opt} = \operatorname*{argmin}_{d \in [d_{\min}, d_{\max}]} d \quad \text{such as:} \quad \mathbb{P}(G(d, \mathbf{X}) < 0) = 1 \tag{2}$$

$$\mathbb{P}(G(d, \mathbf{X}) < 0) = 1, \ \forall d > d_{opt} \tag{3}$$

However, even in this case, it can be quite difficult from a computational point of view to verify that the EMC constraints are respected. Moreover, it is probable that the resulting $d_{opt}$ will be too conservative. For this reason, the probability constraint on $\mathbb{P}(G(d_{opt}, \mathbf{X}) < 0)$ can be relaxed so that if the probability to be EM compatible at the distance $d_{opt}$ is greater than $1 - \alpha$, then we consider that EMC constraints to be respected. The value $\alpha$ is defined by the end-user and is typically set to 0.05 in practice. The optimisation problem formulation we evaluate here is now defined by the following equations:

$$d_{opt} = \operatorname*{argmin}_{d \in [d_{\min}, d_{\max}]} d \quad \text{such as:} \quad \mathbb{P}(G(d, \mathbf{X}) < 0) > 1 - \alpha \tag{4}$$

$$\mathbb{P}(G(d, \mathbf{X}) < 0) > 1 - \alpha, \ \forall d > d_{opt} \tag{5}$$

The sought distance $d_{opt}$ is the lowest distance so that the bundles $A$ and $B$ are EM compatible with probability $1 - \alpha$ and for any distance $d$ greater than $d_{opt}$, the bundles $A$ and $B$ are still EM compatible with a probability at least equal to $1 - \alpha$.

## 4. Sampling and Surrogate Modelling Joint Approach

The proposed algorithm involves an optimisation and the verification of a probability constraint. The objective function of Equation (4) is trivial even if the EMC constraint is not monotonic with $d$. Classical optimiser-under-constraints techniques are fully adapted to the proposed case. For this purpose, we considered in this article constrained optimisation by linear approximation (COBYLA) [6]. COBYLA is a gradient-free optimisation algorithm capable of handling nonlinear inequality constraints. The COBYLA algorithm is a sequential trust-region algorithm that applies linear approximations to the objective and constraint functions. The initialisation value of $d$ in COBYLA is easy to choose by the constraint $\mathbb{P}(G(d, \mathbf{X}) < 0) > 1 - \alpha, \ \forall d > d_{opt}$ as it imposes to start the optimisation algorithm at $d = d_{\max}$.

COBYLA thus performs the optimisation process, and in the meantime, for each value $d$ proposed by COBYLA, the EMC constraint has to be evaluated in order to guide the optimisation.

### 4.1. Classical Monte Carlo

The difficulty of Equation (5) lies in the verification of the EMC constraint $\mathbb{P}(G(d, \mathbf{X}) < 0) > 1 - \alpha$ for a given distance $d$. This probability can in fact be estimated in the following way with the Monte Carlo method (MC) [7]:

$$\mathbb{P}(G(d, \mathbf{X}) < 0) \approx \widehat{\mathbb{P}_d^{MC}} = \frac{1}{N} \sum_{i=1}^{N} \mathbf{1}_{G(d, \mathbf{X}_i) < 0} \tag{6}$$

where $\mathbf{X}_i$ are independent and identically distributed (i.i.d.) random variables with distribution $f_{\mathbf{X}}$ and $\mathbf{1}_{G(d, \mathbf{X}_i) < 0}$ is the indicator function that is equal to one if $G(d, \mathbf{X}_i) < 0$ and zero otherwise. Monte Carlo methods provide also a confidence interval with the central limit theorem to quantify the uncertainty of Monte Carlo probability estimate $\widehat{\mathbb{P}_d^{MC}}$. The confidence interval $IC_{0.95}(\widehat{\mathbb{P}_d^{MC}})$ with a 95% confidence level is then given by:

$$\mathbb{P}\Big(\mathbb{P}(G(d,\mathbf{X}) < 0) \in IC_{0.95}(\widehat{\mathbb{P}_d^{MC}})\Big) = 0.95 \tag{7}$$

with:

$$IC_{0.95}(\widehat{\mathbb{P}_d^{MC}}) = \widehat{\mathbb{P}_d^{MC}} \pm 1.96\sqrt{\frac{\widehat{\mathbb{P}_d^{MC}}(1 - \widehat{\mathbb{P}_d^{MC}})}{N}} \tag{8}$$

The value 1.96 approximates the 0.95 quantile of a standard Gaussian random variable. It can be generalised in the following way: the confidence interval $IC_\gamma(\widehat{\mathbb{P}_d^{MC}})$ with a $\gamma$ confidence level depends on the $\gamma$ quantile of a standard Gaussian random variable. The probability $\mathbb{P}(G(d,\mathbf{X}) < 0)$ of Equation (4) does not need to be estimated too accurately during the optimisation process as many calls to the $G$ function could then be required. In order to adjust the value of $N$ for the estimation of the probabilities $\mathbb{P}(G(d,\mathbf{X}) < 0)$ for different values of $d$, one can consider the confidence interval $IC_{0.95}(\widehat{\mathbb{P}_d^{MC}})$. At least $N = N_{\min}$ MC samples have to be generated to obtain a first estimation of this confidence interval. If $1 - \alpha \notin IC_{0.95}(\widehat{\mathbb{P}_d^{MC}})$, the MC sampling process can be stopped as we can be confident in the decision concerning the EMC at distance $d$. Otherwise, if $1 - \alpha \in IC_{0.95}(\widehat{\mathbb{P}_d^{MC}})$, this means new MC samples are required to reduce the size of $IC_{0.95}(\widehat{\mathbb{P}_d^{MC}})$ as long as the maximum number of MC samples $N_{\max}$ is not reached. If $N = N_{\max}$, then a decision on the constraint respect is made based on the estimation $\widehat{\mathbb{P}_d^{MC}}$ with $N = N_{\max}$. The complete algorithm with MC sampling to evaluate the probability constraint is given in Algorithm A1 for a given distance proposed by COBYLA.

### 4.2. Surrogate Model Strategy

The EMC probability with Monte Carlo sampling presented in the previous section may require many calls to the costly $G$ function. An idea to reduce this cost is to learn a surrogate model $\widehat{G}$ of the $G$ function from an input–output $n$-sample of this function and then estimate the constraint probability on the surrogate model $\widehat{G}$ instead of the $G$ function. If the surrogate model $\widehat{G}$ is not accurate enough on some regions of the input space, it is possible to improve it by calling the true $G$ function for some relevant input positions. The most well-known strategy for probability estimation with the surrogate model and Monte Carlo sampling is notably active learning reliability method combining Kriging and Monte Carlo simulation (AK-MCS) [8], and various evolutions of this algorithms have been proposed recently [9–11]. The probability estimation with a surrogate model relies mainly on four elements:

- The type of surrogate model. Throughout the article, the surrogate model $\widehat{G}$ is assumed to be a conditioned Gaussian process (GP) $\mathcal{G}_n$. Hence, the distribution $\mathcal{G}_n$ knowing the $n$ input–output observations $\{(d,x)_{doe} = ((d,x_1), \ldots, (d_1,x_n)), y = G(d_n, x_{doe})\}$ is Gaussian $\mathcal{G}_n = \mathcal{G}|((d,x_{doe}, y) \sim GP(\mu_n(\cdot), \sigma_n^2(\cdot))$. The initialisation of the design of experiments (DOE) is often performed with Latin hypercube sampling [12]. The mean $\mu_n$ and the variance $\sigma_n^2$ are estimated from $((d,x)_{doe}, y)$. The mean $\mu_n$ is an approximation of $G$, whereas the term $\sigma_n^2$ evaluates the surrogate model error;
- The sampling approach to estimate the probability with the surrogate model. In this article, we only considered Monte-Carlo-based sampling approaches. The constraint probability is estimated with:

$$\mathbb{P}(G(d,\mathbf{X}) < 0) \approx \mathbb{P}(\mu_n(d,\mathbf{X}) < 0) \approx \widehat{\mathbb{P}_d^{MC}} = \frac{1}{N}\sum_{i=1}^{N} \mathbf{1}_{\mu_n(d,\mathbf{X}_i) < 0} \tag{9}$$

where $\mathbf{X}_i$ are i.i.d. samples with distribution $f_\mathbf{X}$. An inferior bound of the probability estimate can be estimated with:

$$\mathbb{P}(\mu_n(d,\mathbf{X}) - 1.96\sigma_n(d,\mathbf{X}) < 0) \approx \widehat{\mathbb{P}_{d,\min}^{MC}} = \frac{1}{N}\sum_{i=1}^{N} \mathbf{1}_{\mu_n(d,\mathbf{X}_i) - 1.96\sigma_n(d,\mathbf{X}_i) < 0} \tag{10}$$

and in the same way, a superior bound is given with:

$$\mathbb{P}(\mu_n(d, \mathbf{X}) + 1.96\sigma_n(d, \mathbf{X}) < 0) \approx \widehat{\mathbb{P}_{d,\max}^{MC}} = \frac{1}{N}\sum_{i=1}^{N}\mathbf{1}_{\mu_n(d,\mathbf{X}_i)+1.96\sigma_n(d,\mathbf{X}_i)<0} \tag{11}$$

- The surrogate model enrichment criterion to properly enrich the surrogate model in order to achieve an accurate approximation of the probability. In this article, the enrichment criterion is the expected feasibility function $EFF(\mathbf{x})$, initially coming from the efficient global reliability analysis (EGRA) method [13] and is given by the following expression:

$$
\begin{aligned}
EFF(d, \mathbf{x}) = \mu_n(d, \mathbf{x}) &\left[ 2\Phi\left(-\frac{\mu_n(d,\mathbf{x})}{\sigma_n(d,\mathbf{x})}\right) - \Phi\left(-\frac{\epsilon + \mu_n(d,\mathbf{x})}{\sigma_n(d,\mathbf{x})}\right) - \Phi\left(\frac{\epsilon - \mu_n(d,\mathbf{x})}{\sigma_n(d,\mathbf{x})}\right)\right] \\
- \sigma_n(d, \mathbf{x}) &\left[ 2\phi\left(-\frac{\mu_n(d,\mathbf{x})}{\sigma_n(d,\mathbf{x})}\right) - \phi\left(-\frac{\epsilon + \mu_n(d,\mathbf{x})}{\sigma_n(d,\mathbf{x})}\right) - \phi\left(\frac{\epsilon - \mu_n(d,\mathbf{x})}{\sigma_n(d,\mathbf{x})}\right)\right] \\
+ \epsilon &\left[ \Phi\left(\frac{\epsilon - \mu_n(d,\mathbf{x})}{\sigma_n(d,\mathbf{x})}\right) - \Phi\left(-\frac{\epsilon + \mu_n(d,\mathbf{x})}{\sigma_n(d,\mathbf{x})}\right)\right]
\end{aligned} \tag{12}
$$

where $\Phi(\cdot)$ is the standard normal cumulative distribution function and $\phi(\cdot)$ the standard normal density function. In EGRA, the expected feasibility function is built with $\epsilon = 2\sigma_n$. At each iteration, the next best point to evaluate $G$ to improve the Gaussian process $\mathcal{G}_n$ is then the candidate sample whose $EFF$ value is maximum among the MC samples generated for probability estimation. The learning stopping condition is based on a stopping value of the learning criterion and is defined as $\max_x(EFF(d, \mathbf{x})) \leq 0.1$ in this article;

- The probability stopping criterion is set to determine when the surrogate model learning is sufficient to obtain an accurate decision on the achievement of the EMC constraint. As long as $1 - \alpha$ belongs to the interval $[\widehat{\mathbb{P}_{d,\min}^{MC}}, \widehat{\mathbb{P}_{d,\max}^{MC}}]$, the surrogate model is not accurate enough to decide if the EM constraint is respected at a distance $d$.

If the learning criterion and the probability stopping criterion are validated at the same time, a decision can be made on the achievement of the EMC constraint between the two bundles at distance $d$. The whole procedure is described in Algorithm A2. The required number of calls to the $G$ function is significantly decreased if compared to the classical MC approach of the previous section. Nevertheless, we also introduced $n_{switch}$ maximum number of calls to $G$ for the evaluation of the constraint at distance $d$ with a Gaussian process. Indeed, the $G$ function can be highly nonlinear notably for high-frequency bands $F$ for which the frequencies have to be considered. A surrogate model is then not adapted, and the risk of misestimation of the EMC probability is not negligible. For this purpose, if the number of iterations becomes larger than $n_{switch} = 250$, this means that the surrogate model is not able to mimic the $G$ function well at the distance $d$, and we propose then to switch to Algorithm A1, taking into account calls to the $G$ function already performed during Algorithm A2. The general scheme of Algorithm A2 is available in Figure 2. The block indicating "call to the $G$ function" implies the launch of the CRIPTE computer code simulation to evaluate the induced currents for a given configuration of cable bundle installation parameters (distance, heights, length).

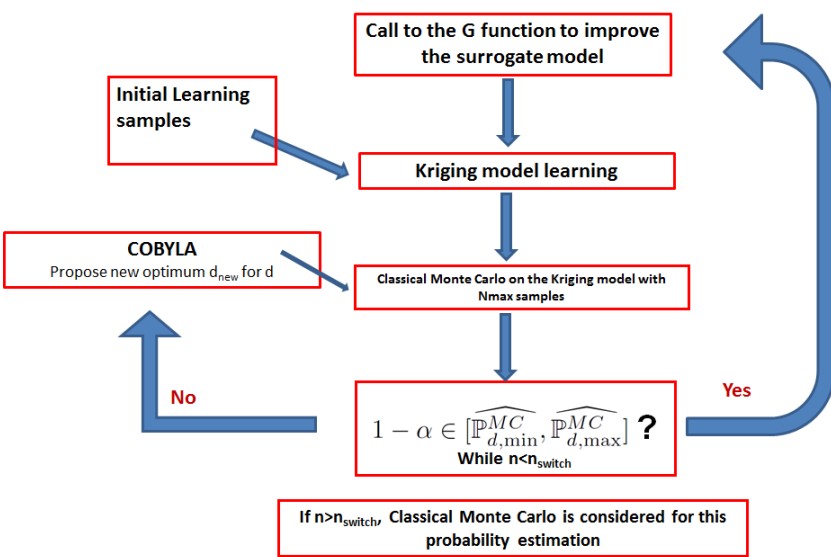

**Figure 2.** Implementation of the COBYLA/Kriging algorithm.

## 5. Application to a Realistic Test-Case

In this section, we propose to apply the two algorithms detailed in Section 4 to a realistic test-case of electric cable bundle segregation.

### 5.1. Description of the Use-Case under Study

We considered in the article the reference test-case schemed in Figure 3. It has the following EM characteristics:

- Bundle $A$ is made of $n_A = 12$ elementary conductors and bundle B of $n_B = 24$ elementary conductors;
- The two bundles were parallel with each other and had the same length;
- All conductors of both bundles were loaded by a 9 $\Omega$ common mode resistance at both extremities;
- A 115 V voltage generator, constant over the frequency range $F = [f_{min}, f_{max}]$, was applied on all elementary conductors of bundle $A$;
- The susceptibility level of all conductors of bundle $B$ was adjusted in $F$ to obtain an optimised segregation distance $d_{opt}$ between both bundles of about 15 cm.

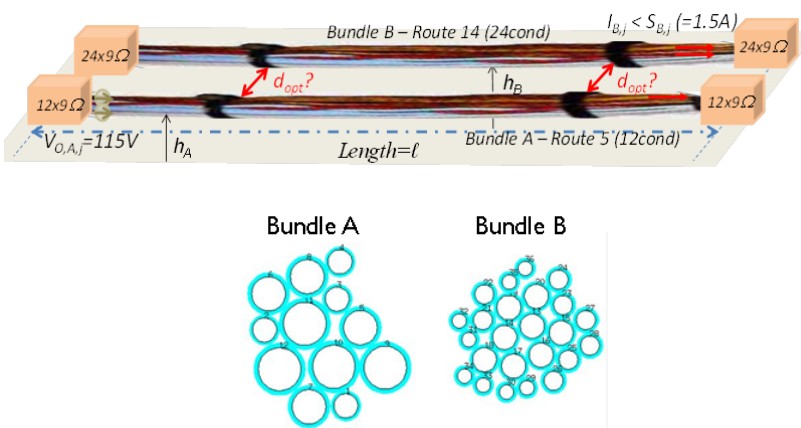

**Figure 3.** Analysis of the EMC between bundle $A$ and bundle $B$.

This use-case is reasonably balanced in terms of the representativity of real industrial cases and the computing resources needed to calculate the G function defined in Equation (1).

Consequently, the objective is to compute $d_{opt}$, for heights of bundles varying between 1.5 cm and 20 cm and a bundle length varying between 6 m and 15 m. This optimum distance should ensure that the probability to be susceptible is lower than $\alpha = 0.05$.

In order to demonstrate the mathematical challenge of this use-case, the current induced on one elementary conductor of bundle B and computed by the CRIPTE computer code [4] is plotted versus frequency in Figure 4 for two very close installation configurations:

- $d = 0.49$ m, $h_A = 9.2$ cm, $h_B = 4.5$ cm, $l = 53$ m;
- $d = 0.49$ m, $h_A = 8.7$ cm, $h_B = 4.0$ cm, $l = 52.8$ m.

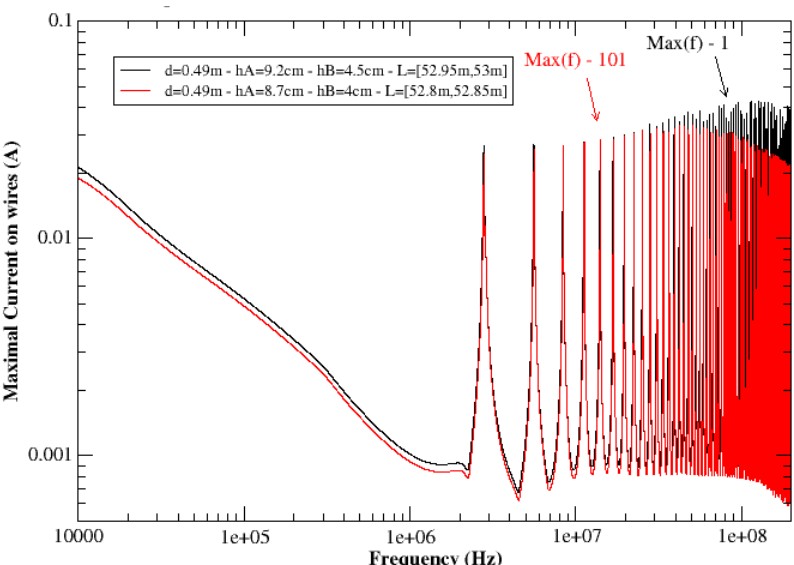

**Figure 4.** Examples of currents induced on bundle *B*.

Figure 4 clearly illustrates the fact that below the resonance frequencies, the current response varies slowly with the installation parameters (distance, heights, and length), and the strategy of optimisation, including the surrogate model, will easily converge. On the contrary, in the domain of the resonance frequencies, we can expect a slow convergence and the need to switch from the surrogate model to the standard Monte Carlo method.

*5.2. Optimisation Strategy Results*

The two proposed algorithms presented in Section 4 were implemented to estimate the optimal segregation distance under the EMC constraint. The algorithms were validated for various configurations of the use-case, chosen in terms of increasing complexity. The validation process consisted of comparing the value of $d_{opt}$ obtained by both algorithms to the value obtained by a brute force Monte Carlo analysis, for which the distance parameter was varied step by step. Figure 5 illustrates the variation of the *G* function for more than 1000 computed geometrical configuration samples with the brute force analysis in a narrow frequency band below resonances $F = [1.99; 2.066]$ MHz and for fixed heights of the bundles. The *G* function is distributed in well-defined strips and can be considered as a monotonic decreasing function with the distance of segregation.

Table 1 summarises the results of the optimisation algorithm compared to the brute force analysis and validates the implementation of the algorithms. The gain in terms of the number of calls to CRIPTE, that is to say in terms of the computational resources to solve this problem, is obvious, and the optimal segregation distance estimation was very well estimated by the proposed algorithm. The linearity of the *G* function facilitates its approximation by a surrogate model. Thus, very few calls to *G* are required to obtain an accurate evaluation of the probability $\widehat{\mathbb{P}_d^{MC}}$ for different values of *d*.

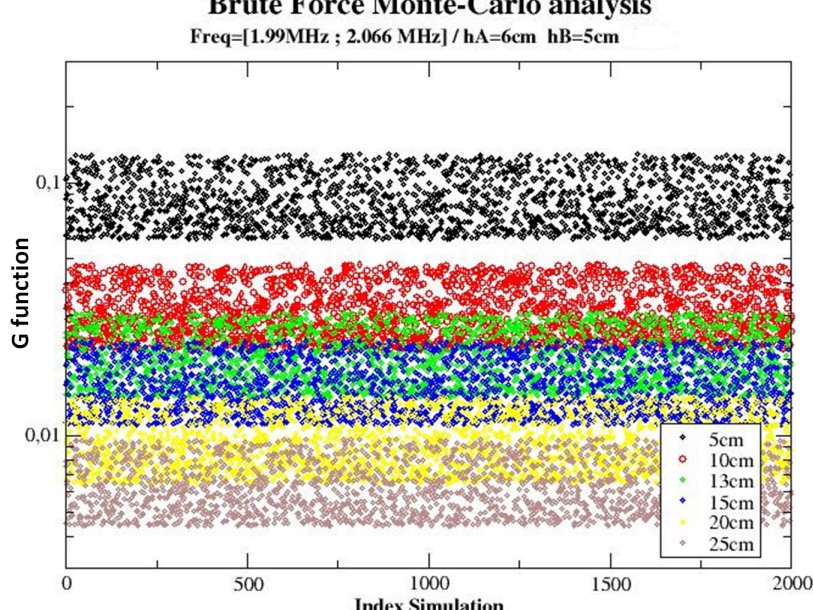

**Figure 5.** G function distribution: brute force analysis $F \approx 2$ MHz and fixed heights of the bundles.

**Table 1.** Brute force, COBYLA/classical MC, and COBYLA/Kriging methods with $F \approx 2$ MHz and fixed heights of the bundles.

|  | Brute Force Monte Carlo | COBYLA/Classical MC | COBYLA/Kriging |
|---|---|---|---|
| $d_{opt}$ | 15 cm | 15 cm ($\pm 1$ m) | 16 cm ($\pm 1$ m) |
| number of calls to $G$ | 12,000 | 1592 ($\pm 100$) | 72 ($\pm 5$) |
| computational time | 160 h | 22 h | 1 h |

The second configuration was much more complex since the analysis was carried out on a larger frequency band [75–125 MHz], which includes several resonance frequencies (see Figure 4); meanwhile, the three installation parameters (length and heights) were varied. Figure 6 shows the variation of the G function for several bundle segregation distances. This time, the strip structure observed in Figure 5 is no longer present and presents a blurred distribution, revealing a non-monotonic variation of the *G* function.

As noted in Table 2 and as expected in this case, the COBYLA/Kriging algorithm needed sometimes to switch to the classical Monte Carlo method in order to converge towards the optimal segregation distance since the *G* function is highly nonlinear and non-stationary. Consequently, the number of iterations of COBYLA/Kriging compared to the first case was much larger, but was still lower than the classical MC approach.

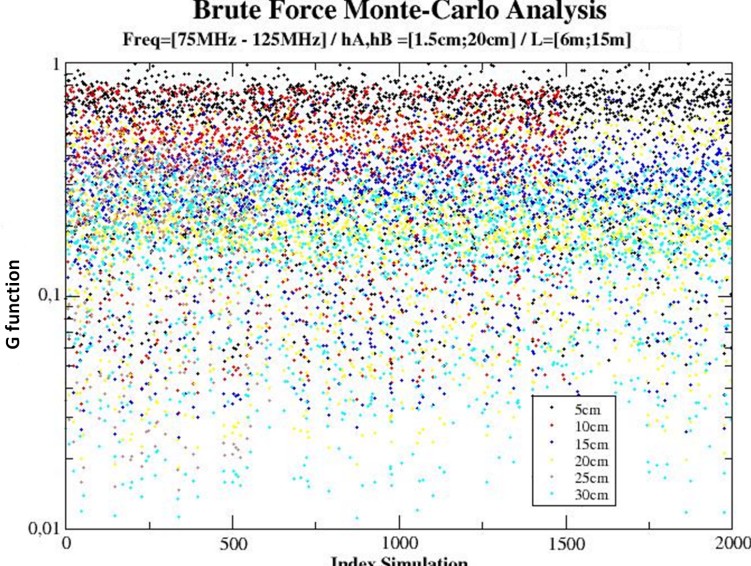

**Figure 6.** *G* function distribution: brute force analysis with $F = [100 \pm 25]$ MHz, heights and lengths varying.

**Table 2.** Brute force, COBYLA/classical MC, and COBYLA/Kriging methods with $F = [100 \pm 25]$ MHz, heights and lengths varying.

|  | Brute Force Monte Carlo | COBYLA/Classical MC | COBYLA/Kriging |
|---|---|---|---|
| $d_{opt}$ | 30 cm | 30 cm ($\pm 1$ m) | 32 cm ($\pm 1$ m) |
| number of calls to *G* | 12,000 | 1304 ($\pm 200$) | 501 ($\pm 200$) |
| computational time | 250 h | 30 h | 12 h |

## 6. Conclusions

In this article, we developed a strategy to evaluate the optimal segregation distance between two electric cable bundles connecting two equipment to ensure their intercompatibility. We first showed that this problem can be solved as an optimisation under constraints. We then proposed a new algorithm based on Monte Carlo samples with a Kriging surrogate model and the COBYLA optimisation method to find the optimal segregation distance. This approach was then applied on two realistic test-cases of electric cable bundle segregation. With a similar algorithm, one can also evaluate the risk to exceed the susceptibility thresholds for a segregation distance lower than the optimum distance, which will give cable bundle integrators possible justification for derogation. At this point, the process of optimisation was fully validated by brute force Monte Carlo analysis. The next step of this work will consist of comparing the simulated results to experimental results on a bundle installation configuration for which their definition, composition, installation, and electrical characterisation are fully controlled.

The method proposed in this paper is a first step in order to make it one day a real industrial tool to help installation and derogate to predefined installation rules with quantitative justifications. The computer code based on this method and developed in the frame of the ANALYST project has been delivered to SAFRAN E&P for further evaluation on their industrial use-cases. As a perspective, there would be a real interest to assess our method against geometries of increasing complexity with additional varying parameters. We think for example of two bundles of different lengths or no longer parallel with the possibility to crossing each other. Finally, it seems that the mathematical concepts developed in this article could be extended in order to evaluate EMS at the scale of a whole network in order to optimise the design of entire cable harnesses made of several branches of cable bundles.

**Author Contributions:** Conceptualization, J.M., I.J. and J.-P.P.; methodology, J.M., I.J. and J.-P.P.; software, J.M. and S.B.; validation, J.M., I.J., S.B. and J.-P.P.; manuscript writing, J.M. and I.J. All authors have read and agreed to the published version of the manuscript.

**Funding:** This research was funded by Clean Sky 2 Joint Undertaking under grant agreement No. 821128.

**Institutional Review Board Statement:** Not applicable.

**Informed Consent Statement:** Not applicable.

**Data Availability Statement:** Not applicable.

**Acknowledgments:** This project received funding from the Clean Sky 2 Joint Undertaking under the European Union's Horizon 2020 Research and Innovation Programme under Grant Agreement No. 821128. Uncertainty analysis was performed with the open-source Python library OpenTURNS [14].

**Conflicts of Interest:** The authors declare no conflict of interest.

## Appendix A. Algorithm of the Proposed Methods

---

**Algorithm A1** EMC probability evaluation with the Monte Carlo method.

---

1: Setting definition: Define $f_{\mathbf{X}}$, $\alpha$, the minimum number of Monte Carlo samples $N_{\min}$ and the maximum number of Monte Carlo samples $N_{\max}$, the distance $d$ proposed by the optimiser COBYLA

2: Initialisation with $N_{\min}$ samples:

3: Generate $N_{\min}$ independent samples $\mathbf{X}_1, \ldots, \mathbf{X}_{N_{\min}}$ with distribution $f_{\mathbf{X}}$

4: Estimate $\widehat{\mathbb{P}_d^{MC}} = \frac{1}{N_{\min}} \sum_{i=1}^{N_{\min}} \mathbf{1}_{G(d, \mathbf{X}_i) < 0}$

5: Compute $IC_{0.95}(\widehat{\mathbb{P}_d^{MC}}) = \widehat{\mathbb{P}_d^{MC}} \pm 1.96 \sqrt{\frac{\widehat{\mathbb{P}_d^{MC}}(1 - \widehat{\mathbb{P}_d^{MC}})}{N_{\min}}}$

6: **if** $\widehat{\mathbb{P}_d^{MC}} - 1.96 \sqrt{\frac{\widehat{\mathbb{P}_d^{MC}}(1 - \widehat{\mathbb{P}_d^{MC}})}{N}} > 1 - \alpha$ **then** Return "the EMC constraint is respected at distance $d$"

7: **end if**

8: **if** $\widehat{\mathbb{P}_d^{MC}} + 1.96 \sqrt{\frac{\widehat{\mathbb{P}_d^{MC}}(1 - \widehat{\mathbb{P}_d^{MC}})}{N}} < 1 - \alpha$ **then** Return "the EMC constraint is not respected at distance $d$"

9: **end if**

10: Iteration until a decision is possible:

11: Set $N = N_{\min}$

12: **while** $N < N_{\max}$ **do**

13:     Set $N = N + 1$

14:     Generate 1 independent sample $\mathbf{X}_N$ with distribution $f_{\mathbf{X}}$

15:     Estimate $\widehat{\mathbb{P}_d^{MC}} = \frac{1}{N} \sum_{i=1}^{N} \mathbf{1}_{G(d, \mathbf{X}_i) < 0}$

16:     Compute $IC_{0.95}(\widehat{\mathbb{P}_d^{MC}}) = \widehat{\mathbb{P}_d^{MC}} \pm 1.96 \sqrt{\frac{\widehat{\mathbb{P}_d^{MC}}(1 - \widehat{\mathbb{P}_d^{MC}})}{N}}$

17:     **if** $\widehat{\mathbb{P}_d^{MC}} - 1.96 \sqrt{\frac{\widehat{\mathbb{P}_d^{MC}}(1 - \widehat{\mathbb{P}_d^{MC}})}{N}} > 1 - \alpha$ **then** Return "the EMC constraint is respected at distance $d$"

18:     **end if**

19:     **if** $\widehat{\mathbb{P}_d^{MC}} + 1.96 \sqrt{\frac{\widehat{\mathbb{P}_d^{MC}}(1 - \widehat{\mathbb{P}_d^{MC}})}{N}} < 1 - \alpha$ **then** Return "the EMC constraint is not respected at distance $d$"

20:     **end if**

21: **end while**

22: The whole budget is necessary to make a decision:

23: **if** $\widehat{\mathbb{P}_d^{MC}} > 1 - \alpha$ **then** Return "the EMC constraint is respected at distance $d$"

24: **end if**

25: **if** $\widehat{\mathbb{P}_d^{MC}} < 1 - \alpha$ **then** Return "the EMC constraint is not respected at distance $d$"

26: **end if**

---

---

**Algorithm A2** EMC probability evaluation with the surrogate model.

---

1:  Setting definition: Define $f_{\mathbf{X}}$, $\alpha$, $n$-sample $((\boldsymbol{d}, \boldsymbol{x})_{doe}, \boldsymbol{y})$, the maximum number of calls $n_{switch}$ to $G$ to enrich the surrogate model, the distance $d$ proposed by the optimiser COBYLA

2:  Initialisation:

3:  Set $j = 1$

4:  Construction of a GP metamodel $\mathcal{G}_n(\boldsymbol{d}, \boldsymbol{x})$ based on the $n$-sample $((\boldsymbol{d}, \boldsymbol{x})_{doe}, \boldsymbol{y})$.

5:  Generate $N$ independent samples $\mathbf{X}_1, \ldots, \mathbf{X}_N$ with distribution $f_{\mathbf{X}}$
Enrichment of the surrogate model until a decision is possible:

6:  **while** $j < n_{switch}$ **do**

7:  　Set $n = n + 1$ and $j = i + 1$

8:  　The learning function $EFF(\boldsymbol{d}, \boldsymbol{x})$ given by Equation (12) is evaluated on $\mathbf{X}_1, \ldots, \mathbf{X}_N$ to find the best candidate $\boldsymbol{x}^*$ to evaluate for enriching the GP metamodel

9:  　The $G$ function is computed on the sample $(\boldsymbol{d}, \boldsymbol{x}^*)$, and the DOE is enriched with this new point $(\boldsymbol{d}, \boldsymbol{x}^*)$ and $G(\boldsymbol{d}, \boldsymbol{x}^*)$

10:  　Construction of a GP metamodel $\mathcal{G}_n(\boldsymbol{d}, \boldsymbol{x})$ of the performance function $G(\boldsymbol{d}, \boldsymbol{x})$ on the DOE

11:  　Estimate $\mathbb{P}(G(\boldsymbol{d}, \mathbf{X}) < 0) \approx \mathbb{P}(\mu_n(\boldsymbol{d}, \mathbf{X}) < 0) \approx \widehat{\mathbb{P}_d^{MC}} = \frac{1}{N} \sum_{i=1}^{N} \mathbf{1}_{\mu_n(\boldsymbol{d}, \mathbf{X}_i) < 0}$

12:  　Compute $\mathbb{P}(\mu_n(\boldsymbol{d}, \mathbf{X}) - 1.96\sigma_n(\boldsymbol{d}, \mathbf{X}) < 0) \approx \widehat{\mathbb{P}_{d,\min}^{MC}} = \frac{1}{N} \sum_{i=1}^{N} \mathbf{1}_{\mu_n(\boldsymbol{d}, \mathbf{X}_i) - 1.96\sigma_n(\boldsymbol{d}, \mathbf{X}_i) < 0}$
and $\mathbb{P}(\mu_n(\boldsymbol{d}, \mathbf{X}) + 1.96\sigma_n(\boldsymbol{d}, \mathbf{X}) < 0) \approx \widehat{\mathbb{P}_{d,\max}^{MC}} = \frac{1}{N} \sum_{i=1}^{N} \mathbf{1}_{\mu_n(\boldsymbol{d}, \mathbf{X}_i) + 1.96\sigma_n(\boldsymbol{d}, \mathbf{X}) < 0}$

13:  　**if** $\widehat{\mathbb{P}_{d,\min}^{MC}} > 1 - \alpha$ **and** $EFF(\boldsymbol{d}, \boldsymbol{x}^*) < 0.1$ **then** Return :the EMC constraint is respected at distance $d$"

14:  　**end if**

15:  　**if** $\widehat{\mathbb{P}_{d,\max}^{MC}} < 1 - \alpha$ **and** $EFF(\boldsymbol{d}, \boldsymbol{x}^*) < 0.1$ **then** Return "the EMC constraint is not respected at distance $d$"

16:  　**end if**

17:  **end while**

18:  The whole budget is necessary:

19:  **if** $j = n_{switch}$ **then** Switch to Algorithm 1 for the probability estimation

20:  **end if**

---

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
