# Peer review of "Optimisation of Segregation Distances between Electric Cable Bundles Embedded in a Structure"

_applsci, doi:10.3390/app12042132_

Round 1
Reviewer 1 Report
The paper presents a method for finding the optimal segregation distance between cable bundles. Although the Authors recognise that the method is still under development and not an industrial tool yet, the paper is well organised and presented.
Probably more results (more use-cases) would have been appreciated by the readers.
No experimental validation has been carried out.
I have a few minor comments.
All equations should be numbered (please see pp. 5 and 6).
The choice of the constant of 1.96 for the bounds of probability should be justified.
The algorithms can be moved to appendices to ease reading of the paper.
At line 51, please check the name of IDS because it seems not correct. Moreover, please use “University” instead of the French translation.
At line 76, adding “EM” to “self-compatible” would make the concept clearer.
At line 83, it is “criterium” as “criteria” is a plural noun.
At line 89, “of the distance” can be added after “the smallest value” as it seems that “d” has not be defined yet?
At line 99, it should be “lead to EM…”
In fig. 4 dots should be used in decimal numbers.
Reviewer 2 Report
EM compatibility is not commonly used in the literature. Use EMC in the paper. EM susceptibility is not commonly used in the literature. Use EMS in the paper.
In line 68 & Figure 1, change V0,A,j to V0,A,i and I0,A,j to I0,A,i. In line 70, SB,i should be changed to SB,j. In section 2.1 above figure 1, it should be said that both cable bundles are in parallel to each other in the whole path as well. In figure 1, the index j in IA,j, VA,j, and SA,j should be changed to i.
Indicate the ground plane in figure 1.
“2.1 Electric cable-bundle notation” might be changed to the Electric cable-bundle configuration or geometry.
In subsection 2.1, it should be said that the configuration of each bundle is affected on the crosstalk (XT) as well.
The definitions of OA,I and OB,j are not obvious. These voltages or currents are produced due to which sources, VO,A,i or VO,B,j or both together?
The definition of Eq. (1) is not obvious to the readers. Consider a single wire for both parallel cables. We only excite one of the single wires and calculate near-end and far-end crosstalks. Ignoring the losses for both wires, (OA,i – SA,i) is not zeros in the frequency domain. SA,i is a shifted version of OA,i. is it correct? In the frequency domain, the amplitude of both signals is equal. I can not understand. I think Eq. 1 should be updated for amplitudes.
In the frequencies that the amplitude of signals is zeros, what is the meaning of Eq. 1? the denominator will be zeros in these frequencies.
Consider that the excitation signal is a gaussian pulse. How does the author define the frequency range to calculate Eq. 1?
If the frequency of both cables is different, how do you define the EQ. 1?
How Eq. 1 can consider the effects of reflections from both sides of the cables?
G < 0 means that the A and bundles are compatible, why?
Reviewer 3 Report
This is an interesting algorithm based on Monte-Carlo sampling with Kriging surrogate model. It is better that the authors discuss the difference of computational time consumption among different methods.
